# Lateral Spread Response: Unveiling the Smoking Gun for Cured Hemifacial Spasm

**DOI:** 10.3390/life13091825

**Published:** 2023-08-29

**Authors:** Kyung Rae Cho, Sang Ku Park, Kwan Park

**Affiliations:** 1Department of Neurosurgery, Konkuk University Medical Center, Seoul 05030, Republic of Korea; medicasterz@gmail.com (K.R.C.); heydaum@daum.net (S.K.P.); 2Department of Neurosurgery, School of Medicine Sungkyunkwan University, Seoul 16419, Republic of Korea

**Keywords:** lateral spread response, abnormal muscle response, hemifacial spasm

## Abstract

Hemifacial spasm (HFS) is a rare disorder characterized by involuntary facial muscle contractions. The primary cause is mechanical compression of the facial nerve by nearby structures. Lateral spread response (LSR) is an abnormal muscle response observed during electromyogram (EMG) testing and is associated with HFS. Intraoperative monitoring of LSR is crucial during surgery to confirm successful decompression. Proper anesthesia and electrode positioning are important for accurate LSR monitoring. Stimulation parameters should be carefully adjusted to avoid artifacts. The disappearance of LSR during surgery is associated with short-term outcomes, but its persistence does not necessarily indicate poor long-term outcomes. LSR monitoring has both positive and negative prognostic value, and its predictive ability varies across studies. Early disappearance of LSR can occur before decompression and may indicate better clinical outcomes. Further research is needed to fully understand the implications of LSR monitoring in HFS surgery.

## 1. Introduction

Hemifacial spasm (HFS) is a rare neuromuscular disorder characterized by involuntary contractions of the facial muscles, predominantly affecting one side of the face. The underlying pathophysiology of HFS involves mechanical compression of the facial nerve at the root exit zone (REZ) by adjacent structures, including arteries, veins, and tumors. Identification of the offending vessel can be achieved through magnetic resonance imaging and confirmed during surgical intervention. However, in some cases, there may be multiple compressing structures, or they may elude detection during the surgical procedure. Therefore, it is crucial for surgeons to accurately identify the primary cause when manipulating the surface of the brainstem. Intraoperative monitoring can be valuable in confirming successful decompression while minimizing the risk of unnecessary manipulation near delicate structures.

Lateral spread response (LSR), an abnormal muscle response (AMR), represents a distinctive neurophysiological characteristic of HFS that remains undetectable under normal conditions. When one branch of the facial nerve is stimulated, an atypical response is observed on the electromyogram (EMG) in other branches of the facial nerve. This anomalous response was initially reported by Janetta and Moller [1,2], and subsequent studies have been conducted to elucidate its significance and its connection to the pathophysiology of HFS. Although LSR holds substantial importance during microvascular decompression surgery (MVD) for HFS, its precise implications are not yet fully understood. The presence of LSR indicates the existence of aberrant cross-connections between facial nerve branches or fibers, although the exact nature of these abnormal connections remains to be identified [3].

The objectives of this article are to provide a comprehensive review of the pathophysiology of LSR, the appropriate techniques for its accurate monitoring, and its clinical implications, and to address the controversies that have been previously discussed in the literature.

## 2. Methods for Monitoring Lateral Spread Response (LSR)

LSR monitoring is typically conducted throughout the entire surgical procedure, from the initiation of general anesthesia until its conclusion. The monitoring of LSR does not interfere with the surgical process, allowing for continuous observation while manipulating the facial nerve and adjacent vessels. Generally, monitoring of LSR begins after the insertion of electrodes, before dural opening, after dural opening, during REZ decompression, and after dural closure [4]. Even if LSR disappears immediately after dural opening or decompression, its recurrence may indicate unsuccessful decompression. Hence, continuous monitoring of LSR is recommended throughout the surgery, even if it disappears initially.

### 2.1. Anesthesia

Since LSR is an abnormal electromyogram (EMG), appropriate anesthetic agents must be used to avoid interfering with the accurate monitoring of this abnormal muscle response. During the induction of general anesthesia, anesthesiologists typically administer neuromuscular blockade (NMB) agents such as rocuronium or vecuronium to minimize patient stress during endotracheal intubation [5]. However, continuous administration of these agents can affect the precise detection of LSR. Therefore, careful titration of NMB is necessary, and many anesthesiologists prefer to delay its administration until the end of the surgery [6]. It is generally recommended that a train-of-four count of more than two be maintained, for accurate monitoring. However, maintaining partial NMB with a target T1/Tc ratio of 50% has proven to be clinically acceptable for LSR monitoring and surgical conditions during MVD [7]. Complete termination of NMB did not significantly enhance LSR monitoring when compared to maintaining a T1/Tc ratio of 50%; thus, it is not recommended in MVD surgery [8].

Inhalational anesthetics have the potential to inhibit or block LSR [9,10]. Studies have also demonstrated significant alterations in the chronaxie of human corticospinal axons when exposed to the inhalational anesthetic sevoflurane [10]. There are reports indicating that desflurane can suppress LSR amplitude by 43%, compared to total intravenous anesthesia alone [9].

### 2.2. Electrode Position

The accurate positioning of electrodes significantly affects the results of neuromonitoring. Misplaced electrodes may stimulate unintended structures, leading to monitoring artifacts instead of capturing meaningful waves. Therefore, ensuring correct electrode positioning is crucial for the precise monitoring of LSR. Typically, the stimulation electrode is placed between the ipsilateral tragus and the external canthus of the eye, where the zygomatic branches of the facial nerve are located. Since the zygomatic branch of the facial nerve innervates the orbicularis oculi muscle, the abnormal muscle response recorded in other facial muscles, such as the frontalis (temporal), orbicularis oris (buccal), and mentalis (marginal mandibular), is defined as LSR [11]. Some institutions have also explored stimulating the marginal mandibular branch, which is located at the border of the mandible and lateral to the mental tubercle, and recording from the orbicularis oculi muscle and the mentalis muscle [3,5].

The conventional stimulation method involves placing paired dermal electrodes in such a way that the cathode is positioned at the proximal branch and the anode is positioned at the distal branch, resulting in centripetal impulses toward the brainstem. However, Lee et al. [12] conducted a study in which they inverted this method by placing the cathode at the distal branch and the anode at the proximal facial nerve. They found that the innervated muscles responded more sensitively to this new stimulation method. In fact, the new method with the cathode at the distal branch demonstrated a higher detection rate for the disappearance of LSR than that of the conventional method. The conventional method achieved a detection rate of 61.8%, while the new method achieved a detection rate of 98.2%. Additionally, after surgical decompression, the conventional method still showed a remaining LSR rate of 29.1%, while the new method had a remaining LSR rate of only 1.8% (Figure 1).

### 2.3. Stimulation Parameters

Currently, there is no standardized guideline for the stimulation and recording methods used in LSR monitoring. Due to variations in nerve excitability thresholds among individuals and the potential influences of anesthesia and other conditions, establishing specific parameters for stimulation presents challenges. However, in most studies, a pulse wave with a duration of 0.2–0.3 ms and an intensity ranging from 5 to 25 mA have been commonly employed. Within this intensity range, LSR can be consistently detected [11].

Nevertheless, although the results have not yet been published, the authors of this review paper discovered the presence of superficially spreading artifacts that can be mistaken for LSR during high-intensity stimulation. In cases where LSR disappears early or goes undetected, examiners may be inclined to increase the stimulation intensity to reveal any hidden LSR. However, the authors’ study revealed the presence of artifacts that mimic LSR, particularly when the abnormal muscle response appears with a very short latency—specifically, less than 10 ms. Therefore, it is advisable that the stimulation intensity not be increased in such cases, where the abnormal muscle response seems to appear too quickly, as it may be due to these artifact responses (Figure 2).

## 3. Prognostic Value of LSR

Zhang et al. conducted a meta-analysis of 14 papers that investigated the prognostic value of lateral spread response (LSR) during microvascular decompression (MVD) [13]. Their findings indicated that the disappearance of LSR is highly associated with short-term outcomes. However, they did not find a significant predictive effect on long-term outcomes. Another systematic review, by Nugroho [4], also concluded that short-term outcomes are strongly correlated with the resolution of LSR. However, the resolution of LSR does not significantly impact long-term outcomes, as patient outcomes tend to improve over time with adequate decompression, even if LSR persists after surgery.

In contrast, a meta-analysis performed by Thirumala et al. [14] revealed that intraoperative LSR monitoring demonstrates high specificity but low sensitivity in predicting a postoperative hemifacial spasm (HFS)-free status at discharge, at 3 months after discharge, and at 1 year after discharge. According to their analysis, the sensitivity was calculated as 40%, 41%, and 40%, respectively, while the specificity was estimated as 89%, 90%, and 89%, respectively, at discharge, 3 months after discharge, and 1 year after discharge. They further calculated the negative predictive value, which indicates the probability of patients achieving LSR resolution, as 92.7%, 95.8%, and 96.0%, respectively, at discharge, 3 months after discharge, and 1 year after discharge. Additionally, the positive predictive value, representing cases where LSR persists, was determined as 47.8%, 40.8%, and 24.4%, respectively, at discharge, 3 months after discharge, and 1 year after discharge. These results suggest that both short-term and long-term outcomes can be predicted based on the resolution of LSR during surgery. However, it should be noted that even if LSR persists during surgery, long-term outcomes may still be positive.

### 3.1. Positive Prognostic Value

Lee et al. [12] reported that the AMR monitoring during MVD is beneficial for identifying the offending vessel and suggesting the most appropriate surgical endpoint. Kong et al. [15] reported that the monitoring of AMR is an effective tool when performing complete decompression, and it may help to predict the outcomes. Some patients still had residual spasm despite LSR disappearance. Various and complicated findings of the offending vessels, as stated in that report, may be the cause of spasm persistence. However, in follow-up visits at 1 year, the number of patients that were included in the category of HFS-free status increased remarkably, so that the correlation between LSR and outcome became significant. Nevertheless, divergent views on this issue have always existed.

Sekula et al. reported that the likelihood of achieving a cure is 4.2 times higher if LSR disappears during surgery than when it persists. However, it is important to note that that meta-analysis only evaluated the utility of LSR at the final follow-up visit and did not consider the postoperative measurements taken two days after surgery [16], Furthermore, there is a consensus among studies that there is a positive relationship between the resolution of LSR and the clinical outcome of HFS. Concerns regarding this relationship arise when there is no LSR observed or when there is early disappearance of LSR. Additionally, questions arise when LSR persists even after successful decompression. In cases where no LSR is observed, or where there is early disappearance, the predictive value becomes uncertain. It becomes challenging to determine the prognosis and the clinical outcome without the presence of LSR as a reliable indicator. Further research is needed to understand the implications and significance of these scenarios.

Similarly, when LSR persists despite successful decompression, the correlation between LSR and clinical outcomes becomes less straightforward. The persistence of LSR may indicate the presence of additional contributing factors or complexities that influence the overall outcome. These cases highlight the multifactorial nature of HFS and the need for a comprehensive assessment of various clinical factors to determine the prognosis and treatment outcomes accurately. However, the resolution of LSR followed by successful decompression of the vessel compressing the facial nerve REZ suggests a positive clinical outcome (Figure 3).

### 3.2. Negative Prognostic Value

While most studies concur that the disappearance of lateral spread response (LSR) is associated with favorable outcomes in hemifacial spasm (HFS), there are some studies that demonstrate a lack of correlation between LSR resolution and clinical outcomes. This discrepancy may be attributed to intraoperative findings of multiple vessel compressions, as well as to the presence of vessels that are not easily visible behind the facial nerve or vessels coursing around the root exit zone without compressing the nerve. These factors can contribute to residual spasms following microvascular decompression (MVD) [15].

In a study by Wei et al. [17], the efficacy of intraoperative auditory brainstem response (AMR) monitoring in improving the outcomes of MVD for HFS was evaluated. However, the findings indicated that intraoperative AMR monitoring did not significantly enhance the efficacy of MVD for HFS, when performed by skilled surgeons.

Notably, studies by Kiya et al. [18] and Yamashita et al. [19] reported a high proportion of patients who exhibited LSR persistence but were free from HFS symptoms. This finding resulted in an insignificant correlation between LSR resolution and HFS relief. It should be acknowledged that these studies may have been limited by small sample sizes. Kiya et al.’s study lacked remaining spasms in both the LSR-disappearance and the persistence groups, rendering their 3 month follow-up analysis inconclusive. Similarly, Yamashita et al. found no significant correlation in the 1 year evaluation.

Indeed, there are studies that have reached different conclusions regarding the predictive value of intraoperative lateral spread response (LSR) monitoring in the outcomes of microvascular decompression (MVD) [20]. One such study, by El Damaty et al. [21], prospectively analyzed 100 patients with hemifacial spasm (HFS) and found that while LSR could guide the appropriate decompression of the facial nerve during MVD, it did not serve as a reliable predictor of postoperative efficacy. Similarly, Hatem et al. [22] observed that all 10 patients in their study achieved clinical cures despite the persistence of LSR during MVD. That finding raised doubts about the practical usefulness of LSR in the context of MVD.

These studies indicated that there is conflicting evidence regarding the predictive value of intraoperative LSR monitoring in MVD outcomes, highlighting the need for further research and consideration of multiple factors in surgical decision-making [3].

In a review conducted by Neves, the abolition of lateral spread response (LSR) and its correlation with clinical outcomes were examined in a group of 32 patients. The study reported a sensitivity of 100% and specificity of 94% in predicting long-term outcomes based on LSR abolition. However, there was no observed relationship between intraoperative LSR changes and relief from hemifacial spasm (HFS) on the first day after surgery.

These findings suggested that LSR abolition may serve as a reliable predictor of long-term outcomes in HFS patients. Additionally, the presence of increased temporal dispersion in the direct response at the stimulated nerve branch could provide valuable insights regarding LSR status, especially in patients with a history of botulinum toxin treatment [23].

Additionally, the clinical course of hemifacial spasm (HFS) after microvascular decompression (MVD) is characterized by high variability. Although many patients experience immediate relief from spasms following MVD, there are instances in which facial spasms continue to persist for several months or even years after surgery, despite the disappearance of LSR. This variability underscores the complex nature of HFS and indicates that factors other than LSR status contribute to the persistence or recurrence of symptoms [24].

### 3.3. Early Disappearance of LSR

Furthermore, it has been observed that the disappearance of lateral spread response (LSR) can occur early in the surgical procedure, even before any vascular decompression of the facial nerve takes place. This early LSR disappearance can be attributed to various factors, such as changes in the dynamics of cerebrospinal fluid (CSF) upon dural opening, CSF drainage, or minimal cerebellar retraction [25]. In some cases, there may be a transient disappearance of LSR followed by its reappearance at a later stage [5,26].

Jiang et al. [5] conducted a retrospective review of 372 patients. Among them, 33 patients exhibited early disappearance of LSR. The study found that the injection of muscle relaxants could diminish LSR and, importantly, that early disappearance of LSR was associated with better clinical outcomes. Several studies have explored the mechanism behind early loss of LSR before decompression in HFS surgery [5,26,27]. This early loss of LSR suggests that the compression force exerted by the offending blood vessels is relatively mild and can be easily influenced by subtle environmental changes, such as CSF egress. Kim et al. [28] suggested that the disappearance of LSR during dural opening or after CSF drainage, prior to decompression, was correlated with poorer outcomes. They emphasized the importance of surgeons carefully identifying the exact offending vessels in order to optimize surgical outcomes. These findings highlight the dynamic nature of LSR changes during HFS surgery and the potential significance of early LSR disappearance as a predictor of surgical outcomes. Surgeons should be attentive to the timing and patterns of LSR changes to improve identification of the responsible vessels and to optimize treatment strategies.

Figure 4 presents a flow chart that outlines the surgical decision-making process with LSR monitoring. This flow chart provides a visual representation of the sequential steps involved in making informed surgical interventions based on the observed LSR patterns.

### 3.4. LSR Monitoring in Secondary HFS

LSR constitutes an electrophysiological manifestation that arises not only from neurovascular conflicts, but also from influences encompassing the REZ and, potentially, other unidentified mechanisms. Consequently, LSR can manifest within secondary HFS, where lesions exert an impact along the trajectory of the facial nucleus and the facial nerve REZ.

However, within the context of tumors originating from the facial nerve, such as facial nerve schwannomas, reports of facial spasm presentation are conspicuously absent [29,30,31]. This discrepancy may be attributed to the propensity of facial nerve schwannomas to originate, predominantly, from the peripheral aspect of the facial nerve rather than from its intracranial segment. Consequently, the presence of abnormal electrophysiological signs such as LSR are not evident in such cases [32].

## 4. Pathophysiology of LSR

The exact pathophysiology of HFS remains unclear. While the most common cause is believed to be mechanical compression of the facial nerve at the REZ by an adjacent artery, there are cases where venous compression, or no specific vessel near the REZ, is identified. Thus, a more comprehensive explanation is needed beyond the direct compression of the REZ. Due to the vague nature of the disease’s pathophysiology, the physiology of the AMR is also a subject of debate.

### 4.1. Peripheral Theory

The peripheral theory of HFS suggests an abnormal cross-transmission of facial nerve fibers at the site of vascular compression, known as the ephaptic transmission of neural impulses between different branches of the facial nerve [33]. Yamashita et al. [34] conducted a study using double stimulation of the AMR in 12 HFS patients to explain the pathophysiology of the condition. Their results showed a refractory period of 3.4 msec between stimulations, which remained constant within the same patients. This finding provided evidence of a peripheral mechanism underlying the pathophysiology of HFS, in contrast to the central nucleus theory. It indicated that the amplitude and latency intervals of the AMR stimulated at different time sequences remain consistent, suggesting no influence from the facial motor neuron. If the site of abnormal cross-transmission were in the facial nucleus, lateral spread responses would exhibit variable latency and amplitude, similar to the F wave [34]. Wilkinson used a strength–duration analysis to suggest that the AMR is likely mediated by antidromic afferent signaling along the facial nerve, as the chronaxies determined for the AMR and M wave were virtually identical [35].

### 4.2. Central Theory

The hyperexcitability of the facial nucleus has been proposed as a significant contributing factor in the pathogenesis of HFS, as suggested by Moller and Jannetta [36] and Poignonec et al. [37]. Yamakami et al. [38] suggested that the kindling-like hyperactivity of the facial nucleus induced by chronic electrical stimulation is the cause of the AMR. Several reports support the central theory, suggesting that the hyperexcitability of the facial nucleus is the origin of the lateral spread response. Ishikawa et al. [39,40,41] examined F waves pre- and postoperatively and during surgery, finding support for this central theory. By correlating F/M-wave amplitude ratios with lateral spread response F/M-wave amplitude ratios, they concluded that the origin of enhanced F waves is the same as that of the lateral spread response. The F wave exhibits variable latency and amplitude due to the hyperexcitability of the central nucleus [34,42].

## 5. Further Research

Thirumala and colleagues [14], in their meta-analysis, proposed that the definition of lateral spread response (LSR) resolution varies across different studies. They suggested that a prospective international multicenter study, with standardized and established definitions of LSR resolution, would yield more accurate and precise results regarding its prognostic value. This highlights the importance of having consistent criteria for determining LSR resolution in future research, to enhance the comparability and the reliability of findings.

Additionally, Kim’s report [28] emphasized that the timing of LSR resolution may have prognostic significance. To further explore this aspect, more studies comparing the timing of LSR disappearance and its relationship to clinical outcomes should be conducted. Understanding the timing of LSR resolution in relation to patient outcomes can provide valuable insights into the prognostic implications of LSR changes during the course of hemifacial spasm (HFS) and subsequent microvascular decompression (MVD) surgeries.

## 6. Limitations

The primary aim of this paper was to provide a comprehensive overview of the monitoring methods, prognostic values, and pathophysiology of LSR. However, it is important to acknowledge certain limitations that were inherent in the scope and nature of this review. Unlike a traditional systematic review or meta-analysis, this study did not adhere to the conventional structure that includes detailed information about the methodology employed, such as the specific time frame of the literature search, the search terms used, and the inclusion and exclusion criteria. As this review focused on presenting a subjective overview of the topic rather than conducting an exhaustive analysis of the available literature, the absence of these methodological components is recognized. Therefore, it is acknowledged that the level of the evidence in the considered literature was limited, given that it did not fall within the domain of a systematic review or a meta-analysis. Nevertheless, significant efforts were made to maintain a neutral standpoint and to encompass a wide range of concerns related to LSR.

## 7. Conclusions

LSR monitoring has been used to identify the cause of HFS and to guide surgical interventions. This review article discussed the pathophysiology of LSR, the techniques for accurate monitoring, and the clinical implications. The presence and resolution of LSR were found to be associated with short-term outcomes, but their predictive value for long-term outcomes is less clear. Studies have shown conflicting results regarding the correlation between LSR resolution and HFS relief, highlighting the need for further research. Factors such as multiple vessel compressions and vessels that are not easily visible can contribute to residual spasms, even after successful decompression. The early disappearance of LSR before decompression can occur due to various factors. Overall, LSR monitoring is a valuable tool, but further research is needed to fully understand its implications and to optimize its use in HFS treatment.

## Figures and Tables

**Figure 1 life-13-01825-f001:**
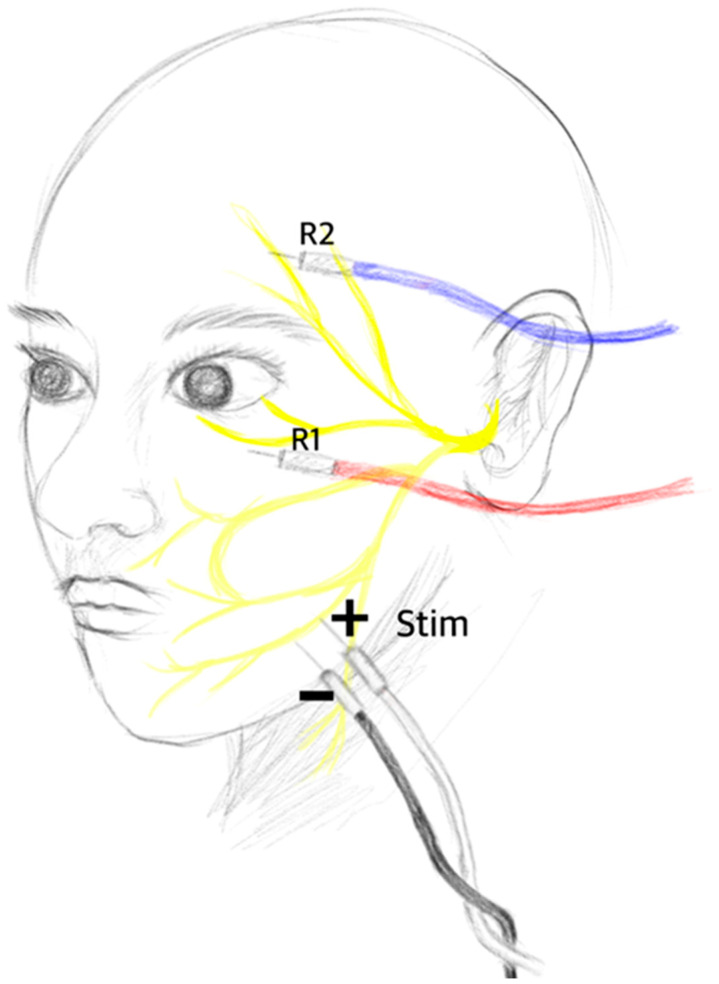
A paired dermal stimulation electrode was positioned either at the zygomatic branch or at the marginal mandibular branch to induce stimulation. Typically, in conventional practice, the cathode is placed proximally while the anode is placed distally to elicit a centripetal impulse. In contrast, Lee et al. conducted a study in which they reversed the placement of the cathode and the anode. Interestingly, this alternative configuration resulted in more sensitive recording of lateral spread response. Their findings suggested that the directionality of the electrode placement can significantly impact the quality and precision of LSR recordings.

**Figure 2 life-13-01825-f002:**
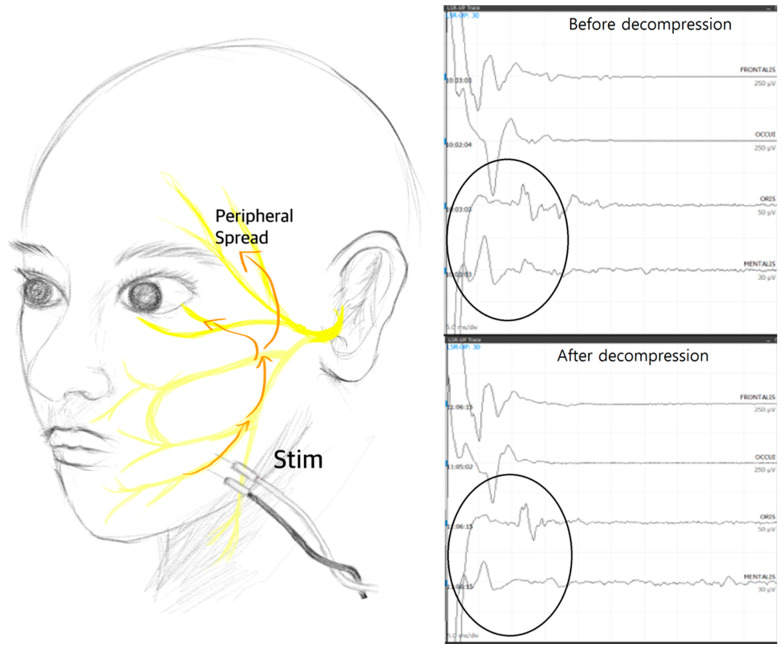
The study conducted by the authors revealed that when the intensity of stimulation surpassed its upper threshold, a direct spreading artifact resembling the low-threshold sensory afferent response (LSR) was observed that did not disappear, even after decompression. This exceeding intensity was quantified by a reduction in the latency of the abnormal muscle response, which was shorter than 10 milliseconds according to their findings. These results suggest that care must be taken to differentiate true LSR from artifacts caused by excessive stimulation intensity, as these artifacts can lead to misleading interpretations and, potentially, affect the accuracy of experimental outcomes.

**Figure 3 life-13-01825-f003:**
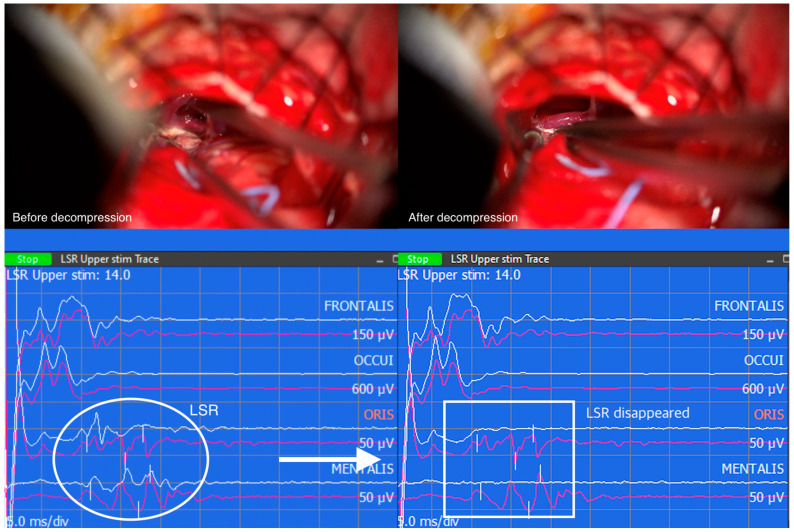
Captured image from intraoperative neuromonitoring program. The offending vessel was dissected from the facial nerve root exit zone, but teflon felt was not yet inserted. As a result, lateral spread response (LSR) is still seen (white oval). After placing teflon felt between the facial nerve and the dissected vessel, LSR disappeared (white rectangle).

**Figure 4 life-13-01825-f004:**
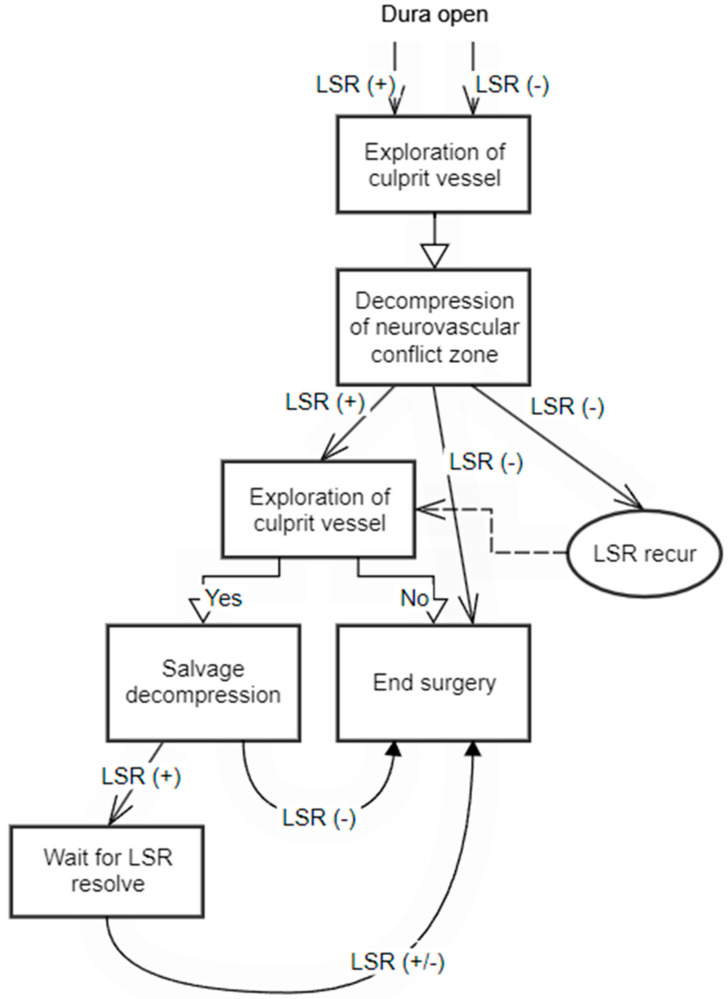
Flowchart for surgical decision-making based on lateral spread response (LSR) monitoring. If LSR disappears before decompression or persists after decompression, careful exploration is warranted. However, the delayed disappearance of LSR is common and, occasionally, early disappearance leads to better outcomes. Therefore, excessive exploration should be avoided to prevent unnecessary intervention.

## Data Availability

Not applicable.

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
