# Peer review of "Lateral Spread Response: Unveiling the Smoking Gun for Cured Hemifacial Spasm"

_life, 2023, doi:10.3390/life13091825_

Round 1

Reviewer 1 Report

The manuscript submitted by Cho et al. made a detailed review of the relationship between LSR and HFS surgery. The review is highly comprehensive, including methodologies with abstractive illustrations, evaluation of prognostic outcomes (both positive and negative), and theories on the pathophysiology of LSR. I consider the review would be highly beneficial for both specialists in peripheral neurosurgery and general neurology.

Author Response

I sincerely appreciate your thoughtful feedback and positive review of my paper. Your insights have provided valuable validation for the effort I have dedicated to this literature. It's encouraging to know that my work has resonated with you and that you found it worthy of commendation.

Reviewer 2 Report

 „The objective of this article is to provide a comprehensive review of the pathophysi-43 ology of LSR“

It is not a proper review it is a subjective overview. For a review, parts of methodology are missing e.g.in which time frame was the literature search, which search terms were used, how many papers were included, how many authors were searching for which terms… etc.

 And the object is not mentioned in the abstract.

Needs to be re- written if it should be submitted as a review.

Minor revision needed

Author Response

Thank you for your insightful feedback on my paper. I appreciate your time and consideration.

I understand your point about the paper not conforming to the typical structure of a comprehensive systematic review or meta-analysis. Your comments about the missing details in the methodology, such as the timeframe of the literature search and search terms used, are duly noted. Additionally, I acknowledge the oversight in not clearly stating the objective in the abstract.

I apologize for these shortcomings and understand the importance of addressing them, given that the paper is intended as a topic review rather than a systematic review or meta-analysis. In the revised manuscript, I will explicitly outline the limitations of not including certain elements expected in systematic reviews and meta-analyses. This will help clarify the paper's focus as a subjective overview of the pathophysiology of LSR.

Your constructive criticism is immensely valuable, and I am committed to making the necessary adjustments to enhance the paper's quality and alignment with its intended purpose as a topic review.

Thank you once again for your guidance.

I put the limitation paragraph above conclusion as below.

          The primary aim of this paper is to provide a comprehensive overview of the monitoring methods, prognostic values, and pathophysiology of LSR. However, it is important to acknowledge certain limitations inherent to the scope and nature of this review. Unlike a traditional systematic review or meta-analysis, this study does not adhere to the conventional structure that includes detailed information about the methodology employed, such as the specific time frame of the literature search, search terms used, and the inclusion and exclusion criteria. As this review focuses on presenting a subjective overview of the topic rather than conducting an exhaustive analysis of the available literature, the absence of these methodological components is recognized. Therefore, it is acknowledged that the evidence level in this literature is limited, given that it does not fall under the domain of a systematic review or meta-analysis. Nevertheless, significant efforts have been made to maintain a neutral standpoint and encompass a wide range of concerns related to LSR.

Reviewer 3 Report

Dear Authors,

Your manuscript describes in detail the mechanisms for central facial nerve compressive pathology.

However, there are some aspects that require improvement.

Please illustrate the technique for decompression with some intra operative immages, not only with graphic schemes.

In the discussion section expand the differential diagnosis with data about periferal facial nerve compression, such as the case of schwannomas. Reference this to newer article from MDPI by Vrinceanu, D.; Dumitru, M.; Popa-Cherecheanu, M.; Marinescu, A.N.; Patrascu, O.-M.; Bobirca, F. Extracranial Facial Nerve Schwannoma—Histological Surprise or Therapeutic Planning? Medicina 2023, 59, 1167. https://doi.org/10.3390/medicina59061167

At the end of the manuscript insert the author contributions, the ethics and copyright statements.

Looking forward to receiving your improved manuscript. 

Author Response

Thank you for your valuable revision suggestions for my manuscript. I will incorporate intraoperative images depicting the decompression procedure, enhancing the clarity of the technique's presentation. Additionally, I will expand the discussion section to include insights on peripherally related facial nerve compression, exemplified by the case of schwannomas as discussed in the referenced article by Vrinceanu et al. (2023). This will contribute to a more comprehensive review of LSR, aligning with the scope that my literature intends to convey. Your input is greatly appreciated.

I added sentence, figure of surgical image and LSR change during decompression in line 183-190

I added paragraph of LSR monitoring in secondary HFS in line 272-290

I incorporated your suggestion to include a discussion of peripheral nerve schwannoma. However, it should be noted that LSR is not observed in cases of peripheral facial schwannoma, rendering such cases unsuitable for inclusion in this review.

At the end of manuscript, I put author contribution forms, ethics and copyright statements as well.

Comment to the editors of "Life"

Given that the manuscript files have been adapted to their publishing formats, I am unable to make corrections without affecting the established format. While I intended to include references from additional cited sources, the document files have already been finalized with citations in place. Consequently, I have indicated the references I cited by highlighting them in bright pink. I kindly request your assistance in ensuring that the intended forms are adhered to during the revision process.

Round 2

Reviewer 3 Report

Dear Authors, 

You managed to answer all the questions from the reviewers.

But put the references at the end of the manuscript.

Looking forward to working toghether on another manuscript.